# Lipophilic Grape Seed Proanthocyanidin Exerts Anti-Cervical Cancer Effects in HeLa Cells and a HeLa-Derived Xenograft Zebrafish Model

**DOI:** 10.3390/antiox11020422

**Published:** 2022-02-19

**Authors:** Changhong Li, Linli Zhang, Chengmei Liu, Xuemei He, Mingshun Chen, Jun Chen

**Affiliations:** 1State Key Laboratory of Food Science and Technology, Nanchang University, Nanchang 330047, China; 357900210034@email.ncu.edu.cn (C.L.); zll231797@126.com (L.Z.); liuchengmei@ncu.edu.cn (C.L.); chenjun@ncu.edu.cn (J.C.); 2Agro-Food Science and Technology Research Institute, Guangxi Academy of Agricultural Sciences, Nanning 530007, China; xuemeihegx@gxaas.net; 3Guangxi Key Laboratory of Fruits and Vegetables Storage-Processing Technology, Nanning 530007, China

**Keywords:** lipophilic grape seed proanthocyanidin, cervical cancer, HeLa cell, apoptosis, zebrafish

## Abstract

Lipophilic grape seed proanthocyanidin (LGSP) synthesized from GSP and lauric acid exhibits an excellent antioxidant and anti-inflammatory effect. However, its anti-cervical cancer activity is still unknown. In this study, the in vitro anti-cervical cancer activity of LGSP on HeLa cell lines was investigated by MTT assay, flow cytometry and Western blot analysis, and its effect was explored by a HeLa-derived xenograft zebrafish model. LGSP exhibited an excellent anti-proliferative effect on HeLa cells by increasing the level of reactive oxygen species, which further induced cell apoptosis and blocked cell cycle progression in the G2/M phase. LGSP-treated HeLa cells showed a reduction in mitochondrial membrane potential, upregulation of the Bax/Bcl-2 ratio, release of cytochrome *c* into the cytoplasm, and activation of cleaved caspase-9/3 and cleavage of PARP, thus indicating that LGSP induced apoptosis through the intrinsic mitochondrial/caspase-mediated pathway. In the zebrafish model, LGSP effectively suppressed the growth of a HeLa xenograft tumor. These data suggest that LGSP may be a good candidate for the prevention or treatment of cervical cancer.

## 1. Introduction

Cervical cancer is the fourth most common cancer in women with approximately 570,000 new cases and 311,000 deaths reported worldwide in 2018 [1]. Although significant progress has been made in the treatment of cervical cancer, they are costly and unbearable. The prevention of cancer through vaccination has become very important [2], and increasing attention has been paid to its health effects. A large number of epidemiologic studies have investigated the relation between dietary polyphenols and cancer [3], and these have demonstrated that phenolic compounds can reduce the risk of chronic diseases, including cancer, diabetes, cardiovascular and neurodegenerative diseases [4,5].

Phenolic compounds, which are widely distributed in food, are regarded as an important source of antioxidants in the daily diet. Grape seed proanthocyanidin (GSP), the major polyphenol component in grape seed, has been widely used as a dietary supplement and food additive due to its various biological activities [6]: antioxidant [7], anti-tumor [8], anti-inflammatory [9] and anti-cardiovascular disease [10]. However, the hydrophilicity of GSP with its low solubility in lipid systems limits its application in lipidic food matrices and hinders its bioactivity [11].

In recent years, lipophenols have received much attention due to their high antioxidant effects on lipidic food matrices [12,13]. Tea polyphenol palmitate has been allowed as a food additive to fat and oils by the China Food and Drug Administration [14]. Lipophilic grape seed proanthocyanidin (LGSP), synthesized from GSP and lauric acid, has shown antioxidant and anti-inflammatory activity higher than that of GSP [15,16]. Chronic inflammation and cancer in organisms are regulated and stimulated by oxidative stress; thus, the good antioxidant and anti-inflammatory activity of LGSP may also have certain anti-cancer activity [17]. However, the anti-cervical cancer effect of LGSP is still unclear.

In this study, we examined the hypothesis that LGSP has an anti-cervical cancer effect on HeLa cells, and we explored the underlying mechanisms by MTT assay, flow cytometry and Western blot analysis. We then probed the anti-cervical cancer effect of LGSP in vivo, which provided scientific evidence for its future application in cervical cancer prevention or therapy.

## 2. Materials and Methods

### 2.1. Materials

GSP (proanthocyanidin content, 95%) was purchased from Xi’an Realin Biotechnology Co. (Xi’an, Shanxi, China). Dulbecco’s modified Eagle medium (DMEM), trypsin/EDTA and trypsin were obtained from Gibco (Grand Island, NY, USA). Fetal bovine serum (FBS) was purchased from Biological Industries (Shanghai, China). Thiazolyl blue tetrazolium bromide (MTT) was purchased from Aladdin Biotechnology (St. Louis, MO, USA). The Cell cycle staining kit and Annexin V-FITC/PI apoptosis kit were obtained from MultiSciences (Shanghai, China). Mitochondrial membrane potential (MMP) assay kit, reactive oxygen species (ROS) assay kit and bicinchoninic acid (BCA) protein assay kit were purchased from Beyotime Biotechnology (Shanghai, China). The antibodies against cleaved PARP, PARP, caspase-3, cleaved caspase-3, caspase-9, cleaved caspase-9, cytochrome *c*, Bax, Bcl-2 and GADPH were purchased from Cell Signaling Technology (Beverly, MA, USA).

### 2.2. Preparation of LGSP

LGSP was prepared as previously described [15]: GSP, lauric acid, immobilized lipase TLIM (Novozymes, Shanghai, China) and ethanol were mixed into a screw-capped glass bottle. After incubating for 12 h with magnetic stirring at 50 °C, the mixture was filtrated to remove the enzyme and stop the reaction. The product was condensed using a rotatory evaporator (RE-52AA, Yarong Co. Ltd., Shanghai, China) to obtain LGSP, the major compositions of which are shown in Appendix A: 3′-*O*-lauroyl catechin, 3′-O-lauroyl epicatechin, 3′,5′-2-*O*-lauroyl epigallocatechin, 3′,3″,5″-3-*O*-lauroyl epicatechin gallate, 3′,3″-2-*O*-lauroyl procyanidin A2,3′,3″-2-*O*-lauroyl procyanidin B1 and 3′, 3″,3′′′-3-*O*-lauroyl procyanidin C1.

### 2.3. Cell Culture

HeLa cells, obtained from the Chinese Academy of Sciences (Shanghai, China), were cultured in DMEM supplemented with 10% fetal bovine serum and 1% penicillin–streptomycin at 37 °C under a 5% CO_2_ humidified atmosphere.

### 2.4. Cell Proliferation Analysis

The effect of LGSP on the proliferation of HeLa cells was measured by MTT assay according to the literature with a slight modification [18]. Briefly, the cells (5 × 10^3^) were seeded into 96-well plates and treated with various concentrations (25, 50, 100, 150 and 200 µg/mL) of LGSP for 24 and 48 h. Then, 0.5 mg/mL MTT solution was added and incubated for 4 h. After that, 200 μL of DMSO was added to dissolve the formazan crystals, and the absorbance was measured at 490 nm by a microplate reader (Thermo Scientific, Waltham, MA, USA).

### 2.5. Cell Apoptosis Analysis

Cell apoptosis was analyzed by flow cytometry using the Annexin V-FITC cell apoptosis kit [18]. HeLa cells (4 × 10^5^) were seeded into 6-well plates and incubated with various concentrations (50, 100 and 200 µg/mL) of LGSP for 48 h. Then, floating and adherent cells were harvested by trypsinization, washed twice with pre-cooling PBS, and resuspended in a 500 µL binding buffer containing 5 µL Annexin V-FITC and 10 µL PI. After incubating for 30 min in the dark at room temperature, the cells were determined by a flow cytometer (BD, Franklin Lakes, NJ, USA). The apoptotic HeLa cells were analyzed by FlowJo v10 software (BD, Franklin Lakes, NJ, USA).

### 2.6. Cell Cycle Distribution Analysis 

The cell cycle distribution was analyzed by flow cytometry as described in the literature [19]. After treatment with LGSP at concentrations of 50, 100 and 200 µg/mL for 48 h, HeLa cells were harvested, washed twice with pre-cooling PBS, and resuspended in 1 mL DNA staining solution with PI and 10 µL permeabilization in the dark at room temperature for 30 min. The DNA content of the stained cells was measured by a flow cytometer (BD, Franklin Lakes, NJ, USA). All results were analyzed using FlowJo v10 software (BD, Franklin Lakes, NJ, USA).

### 2.7. Determination of ROS Production

ROS production was measured using the ROS assay kit with DCFH-DA [20]. The HeLa cells were seeded into 6-well plates at the density of 4 × 10^5^ cells/well. After attachment, the cells were treated with different concentrations (50, 100 and 200 µg/mL) of LGSP for 48 h. Then, they were harvested, incubated with 10 µM DCFH-DH in the dark at 37 °C for 30 min and washed three times with a serum-free medium. ROS production was determined by a flow cytometer (BD, Franklin Lakes, NJ, USA).

### 2.8. Measurement of MMP

MMP was measured according to the instruction of the MMP assay kit with JC-1. HeLa cells (4 × 10^5^) were seeded into 6-well plates. After treatment with different concentrations (50, 100 and 200 µg/mL) of LGSP for 48 h, the cells were obtained by trypsinization, incubated with JC-1 working solution for 30 min in the dark at room temperature, and washed twice with JC-1 buffer. The MMP of HeLa cells was observed under an inverted fluorescence microscope (Nikon Eclipse Ti-U, Melville, NY, USA) and analyzed by a flow cytometer (BD, Franklin Lakes, NJ, USA).

### 2.9. Western Blot Analysis

HeLa cells (4 × 10^5^) were seeded into 6-well plates. After treatment with various concentrations of LGSP (50, 100 and 200 µg/mL) for 48 h, cells were harvested and lysed in a RIPA buffer. The total protein concentration was detected using an BCA protein assay kit [21]. The cell lysates were denatured in a SDS sample buffer and subjected to SDS-PAGE gel. The separated proteins were subsequently transferred onto the polyvinylidene difluoride membrane (Beyotime Biotechnology, Shanghai, China), blocked for 1 h at room temperature, incubated with specific primary antibody (1:1000, *v*/*v*) overnight at 4 °C, and followed by secondary antibody (1:3000, *v*/*v*) for 2 h at room temperature. The target proteins were detected using an ECL kit (Beyotime, Shanghai, China) by a chemiluminescent gel imaging system (Bio-Rad, Hercules, CA, USA).

### 2.10. Zebrafish Maintenance

The zebrafish wild-type AB strain was purchased from the China Zebrafish Resource Center (Wuhan, China). Adult zebrafish were maintained in a recirculating zebrafish housing system with a 14 h light/10 h dark photoperiod at 28 °C. After natural pair-mating and reproduction, the embryos were collected and incubated in fish water at 28 °C (0.2% instant ocean salt, pH 6.9–7.2, conductivity 480–510 μS/cm and hardness 53.7–71.6 mg/L CaCO_3_). The use and manipulations of zebrafish were approved by the ethical review committee of Nanchang University (Nanchang, China) on 25 March 2021.

### 2.11. Embryo Acute Toxicity Test

The acute toxicity test was performed on zebrafish embryos as described in the literature [22]. Briefly, embryos at 10 h post-fertilization (hpf) were randomly distributed to 6-well culture plates (30 embryos/well) and exposed to LGSP with indicated concentrations (4, 8, 16 and 32 μg/mL). The hatched larvae were counted and the death rate was calculated at different time points (24, 48, 72, 96 and 120 hpf).

### 2.12. HeLa-Derived Xenograft Zebrafish Model

The zebrafish xenograft model was established by transplantation of stable fluorescent cervical cells expressing GFP protein (HeLa-GFP) into the yolk sac of zebrafish embryos (48 hpf). An average of 30 zebrafish embryos were used for each treatment. After successful transplantation, the zebrafish embryos were maintained at 35 °C for 1 day to promote tumor-cell proliferation and embryo recovery. After exposure to LGSP (4 and 8 μg/mL) for 48 h, the larvae were placed on agarose, and the fluorescent tumor cells were observed under a Leica KL300 LED inverted fluorescence microscope (Leica, Wetzlar, Germany). The fluorescence intensity of tumor cells was quantified by Image J (Rawak Software, Inc., Stuttgart, Germany) [23].

### 2.13. Statistical Analysis

Data were represented as mean ± standard deviation (SD) from at least three independent experiments. The differences between the experimental and control groups were compared using one-way ANOVA followed by Dunnett’s multiple comparisons test using SPSS software (IBM version 16.0, Chicago, IL, USA). A *p* value < 0.05 was considered statistically significant.

## 3. Results

### 3.1. Effect of LGSP on the Proliferation of HeLa Cells

The anti-proliferative effect of LGSP on HeLa cells was determined using MTT assay. As shown in Figure 1A,B, the inhibition rate of LGSP on HeLa cells increased in a dose-dependent manner at 24–48 h. The inhibition rate increased rapidly from 11.67 to 56.69% after treatment with LGSP at concentrations of 25–50 μg/mL for 48 h, followed by enhancing gradually at concentrations of 50–200 μg/mL. The IC_50_ values of LGSP and GSP for anti-proliferative effect on HeLa cells were 57.97 ± 1.23 and 111.16 ± 1.02 μg/mL, respectively, after 48 h of treatment, indicating that LGSP had stronger anti-cervical cancer activity compared to GSP. The morphology of HeLa cells after treatment with LGSP and GSP at concentrations of 50, 100 and 200 μg/mL for 48 h was shown in Figure 1C. The number of HeLa cells clearly decreased after treatment with LGSP. Meanwhile, HeLa cells became round and lost their characteristic stretched appearance. The morphological changes in HeLa cells treated with LGSP were more obvious than those treated with GSP, which was consistent with the results of the anti-proliferative effect.

### 3.2. LGSP-Induced HeLa Cells Apoptosis

To explore the effect of LGSP on HeLa cell apoptosis, the cells (Annexin V+/PI− fraction subpopulations and Annexin V+/PI+ fraction subpopulations) were detected by flow cytometry. As shown in Figure 2A, both LGSP and GSP displayed a dose-dependent increase in apoptosis induction in HeLa cells. LGSP resulted in 18.77, 31.25 and 43.36% apoptotic rates at concentrations of 50, 100 and 200 μg/mL, respectively, which were higher than that of GSP at the same doses.

### 3.3. Effect of LGSP on the Cell Cycle of HeLa Cells

The effect of LGSP on the cell cycle distribution was analyzed by flow cytometry. As shown in Figure 2B, compared with 18.8% G2/M phase cells of control cells, an increase in the percentage of cells in the G2/M cells (20.1, 24.3 and 36.5%) was observed in cells after treatment with LGSP at concentrations of 50, 100 and 200 μg/mL, respectively. Similar to LGSP, GSP also induced the occurrence of G2/M cell-cycle arrest, but its effect (25.6%) was weaker than that of LGSP at the dose of 200 μg/mL.

### 3.4. LGSP Enhanced the ROS Levels in HeLa Cells

ROS generation plays a vital role in apoptosis [24]. To investigate the effect from LGSP, on the intracellular ROS level, the production of intracellular ROS was examined by flow cytometry. As shown in Figure 3, treatment of HeLa cells with LGSP at concentrations of 50, 100, and 200 μg/mL for 48 h resulted in a significant dose-dependent increase in the fluorescence intensity. Compared with the control group, the fluorescence intensity of HeLa cells treated with LGSP and GSP at the concentration of 200 μg/mL increased 3.01- and 2.12-fold, respectively, indicating the ROS production of HeLa cells after treatment with LGSP was higher than that with GSP.

### 3.5. LGSP Induced Mitochondrial Dysfunction in HeLa Cells

To examine whether LGSP-induced apoptosis in HeLa cells was involved in the mitochondria-mediated pathway, the MMP of HeLa cells treated with LGSP was observed under an inverted fluorescence microscope with fluorescence probe JC-1. The change of JC-1 from red to green fluorescence indicated a decrease in membrane potential. As shown in Figure 4A, red fluorescence intensity decreased while green fluorescence intensity increased, suggesting that LGSP treatment triggered mitochondrial damage. For further quantitative analysis, the fluorescence intensity of HeLa cells was determined by flow cytometry. As shown in Figure 4B, the green fluorescence intensity of HeLa cells treated with 200 μg/mL LGSP increased to 49.22%, which was higher than that for GSP (8.54%).

### 3.6. Effect of LGSP on Bax, Bcl-2 and Cytochrome c Expression in HeLa Cells

To explore the effect of LGSP on apoptosis-regulated proteins, the levels of Bcl-2, Bax and cytochrome *c* on HeLa cells were examined. As shown in Figure 5A, LGSP up-regulated Bax expression and down-regulated Bcl-2 expression, leading to a significant increase in the Bax/Bcl-2 ratio. Moreover, the Bax/Bcl-2 ratio of cells treated with 200 μg/mL LGSP was 29.24, which was higher than that for GSP (8.60) at the same concentration. The Bcl-2 protein family, pro-apoptotic Bax and anti-apoptotic Bcl-2, could strictly control the release of cytochrome *c* from mitochondria into the cytoplasm [25]. The results showed that the level of cytochrome *c* increased in a dose-dependent manner after treatment with LGSP, indicating that LGSP increased the release of cytochrome *c* into the cytoplasm.

### 3.7. LGSP-Induced Activation of Caspases and the Cleavage of PARP Protein in HeLa Cells

The cytochrome *c* released from the mitochondria activated the caspase apoptosis pathway, and we then explored whether LGSP activated it. As shown in Figure 5B, after treatment with LGSP, the levels of cleaved caspase-3, cleaved caspase-9 and cleaved PARP in HeLa cells increased in a dose-dependent manner, indicating that LGSP led to a clear cleavage of caspase-9, caspase-3 and PARP.

### 3.8. LGSP Blocked Cervical Cancer Growth in a HeLa-Derived Xenograft Zebrafish Model

To test whether LGSP can suppress cervical cancer in vivo, we treated zebrafish carrying HeLa xenograft tumors with LGSP. First, we evaluated the toxicity of GSP and LGSP to zebrafish larvae. The results showed that GSP at doses of 4–8 μg/mL and LGSP at doses of 4–16 μg/mL exhibited non-toxic effects (Figure 6A). Hence, the doses of 4 and 8 μg/mL were selected for further study in the HeLa-derived xenograft zebrafish model. As shown in Figure 6B,C, the circled green fluorescence represented the cervical tumor mass, indicating that the HeLa xenograft model was successfully established in zebrafish. LGSP and GSP at doses of 4 and 8 μg/mL markedly decreased the green fluorescence intensity in a dose-dependent manner, and the inhibition rate of LGSP (84.32%) was higher than that for GSP (68.89%) at the concentration of 8 μg/mL.

## 4. Discussion

GSP, the main polyphenol in grape seed, has been proven to inhibit cervical cells by inducing apoptosis and blocking cell cycles [26]. LGSP, synthesized by enzymatic modification of GSP, showed higher antioxidant and anti-inflammatory activity compared to GSP [15,16]. Therefore, we hypothesized that LGSP has a potential chemopreventive effect on cervical cancer.

Our results showed that LGSP exhibited an anti-proliferative effect on HeLa cells, and its effect was higher than that for GSP, which confirmed our hypothesis. Similarly, the oleate derivative of quercetin-3-O-glucoside showed a greater anti-proliferative effect on HepG2 cells compared to quercetin-3-O-glucoside [27]. As we previously reported, the lipophilicity of LGSP was higher than that for GSP [15], which may lead to higher cell membrane permeability and better bioactivity in cell models [12].

Generally speaking, the induction of apoptosis or cell cycle arrest or a combination of these two modes is the common mechanism for inhibiting cancer-cell proliferation [28]. In this study, LGSP increased the apoptosis rate and the percentage of G2/M phase on HeLa cells, indicating LGSP exerted an anti-proliferative effect on HeLa human cervical cancer cells by inducing apoptosis and blocking G2/M cell cycle. It is reported that intracellular ROS was associated with cell apoptosis [29], and could also cause a DNA damage response that could impact the cell cycle progression [30]. In this study, the level of ROS increased dramatically in the HeLa cells treated with LGSP, suggesting that LGSP may induce the production of ROS, leading to cell apoptosis and cycle arrest. In addition, we found that the anti-proliferative effect of GSP on HeLa cells was also achieved by inducing apoptosis and blocking the G2/M cell cycle, which was consistent with the report of Chen et. al [26].

There are two principal apoptosis pathways: the extrinsic pathway and the intrinsic mitochondrial pathway [31], which is the key signaling pathway in the induction of apoptosis [32]. It has been reported that the loss of MMP is a typical phenomenon in mitochondrial-dependent apoptosis [33]. In this work, LGSP decreased the MMP of HeLa cells, indicating that LGSP-induced apoptosis may be through the mitochondrial pathway. The mitochondrial apoptosis pathway is strictly regulated by the two important members of the Bcl-2 protein family: Bax and Bcl-2 [34]. The oligomerization of pro-apoptotic Bax and the reduction of anti-apoptotic Bcl-2 could lead to the release of cytochrome *c* from the mitochondria into the cytoplasm [25]. Our results showed that LGSP clearly increased the Bax/Bcl-2 ratio and promoted the release of cytochrome *c* into the cytoplasm, which confirmed that LGSP induced apoptosis through the mitochondrial pathway.

The cytochrome *c* released from the mitochondria was reported to activate caspase-9 and finally induce apoptotic occurrence [35]. To further study the downstream apoptosis mechanism of LGSP on HeLa cells, we evaluated some relevant proteins of the caspase pathway. Our findings showed that LGSP up-regulated the expression of cleaved caspase-9 caspase-3 and PARP to activate the caspase pathways. The activated cleaved caspases and PARP were considered to be relevant biomarkers for inducing apoptosis [36]. Therefore, our findings suggested that LGSP-induced apoptosis was mediated through the mitochondrial/caspase pathway, which involved the up-regulation of the Bax/Bcl-2 ratio, release of cytochrome *c*, and activation of cleavage of caspase 9, caspase-3 and PARP.

The above results indicated that LGSP has an anti-cervical cancer effect in vitro. It is necessary to evaluate its anti-cervical cancer effect in vivo further. In recent years, zebrafish xenograft models have been generated more frequently to study malignancies due to their special characteristics, such as transparent embryos, large-scale generation, rapid organogenesis and no immune rejection [37,38]. Therefore, we then assessed the effect of LGSP on the growth of HeLa cells in a zebrafish xenograft model. In this study, we found that LGSP remarkably decreased the growth of a HeLa xenograft tumor, and the inhibition rate of LGSP was higher than that for GSP, indicating that LGSP had better anti-cervical cancer activity in vivo, which was consistent with our in vitro studies.

## 5. Conclusions

In summary, this study first indicated that LGSP possessed an anti-cervical cancer effect on HeLa cells in vitro and in vivo. LGSP exhibited an excellent anti-proliferative effect on HeLa cells by inducing cell apoptosis and blocking cell-cycle progression in the G2/M phase. The apoptosis of HeLa cells triggered by LGSP was through the mitochondrial/caspase-mediated pathway, characterized by Bax/Bcl-2 ratio up-regulation, the release of cytochrome *c*, the loss of mitochondrial membrane potential, and the cleaved caspase-9/3 and cleavage of PARP activation. In the zebrafish model, LGSP effectively suppressed the growth of a HeLa xenograft tumor. Furthermore, LGSP presented better anti-cervical cancer effects than GSP in HeLa cells and zebrafish models. LGSP could be considered a potential supplement for preventing and treating human cervical cancer.

## Figures and Tables

**Figure 1 antioxidants-11-00422-f001:**
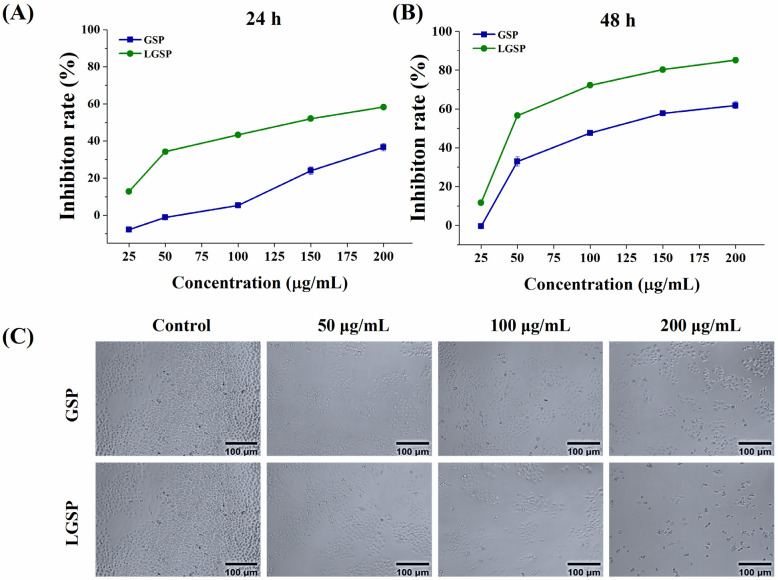
Anti-cervical cancer effect of GSP and LGSP on HeLa cell lines. HeLa cells were treated with various doses of GSP and LGSP for (**A**) 24 h and (**B**) 48 h. (**C**) Morphological image of HeLa cells after treatment with GSP and LGSP for 48 h. Data were mean ± SD (*n* = 3).

**Figure 2 antioxidants-11-00422-f002:**
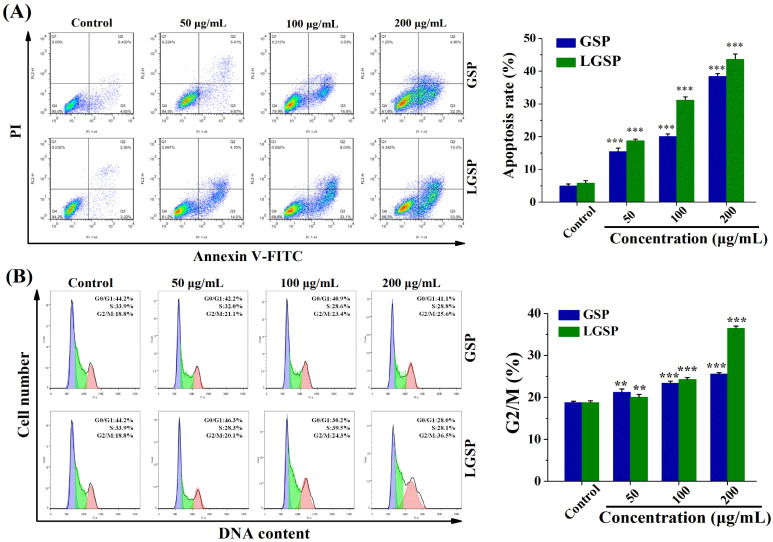
(**A**) Apoptotic effect of GSP and LGSP on HeLa cells double-stained with Annexin V-FITC/PI and analyzed by flow cytometry. (**B**) The cell cycle distribution of HeLa cells treated with GSP and LGSP. HeLa cells were treated with different concentrations of GSP and LGSP for 48 h. Data were mean ± SD (*n* = 3). ** 0.001 < *p* < 0.01; *** *p* < 0.001.

**Figure 3 antioxidants-11-00422-f003:**
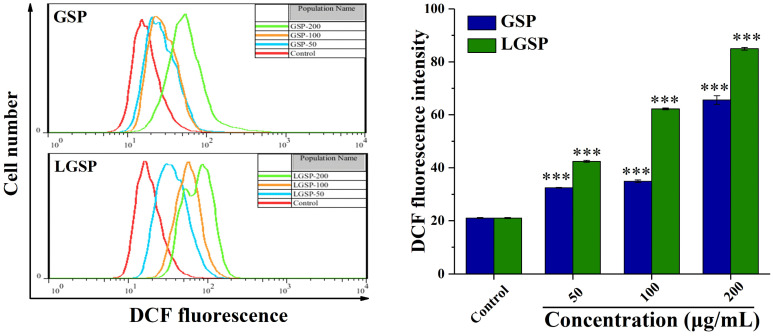
Effect of GSP and LGSP on the ROS generation of HeLa cells treated with different concentrations of GSP and LGSP for 48 h. ROS production was analyzed by flow cytometry using a DCFH-DA probe, and data were mean ± SD (*n* = 3): *** *p* < 0.001.

**Figure 4 antioxidants-11-00422-f004:**
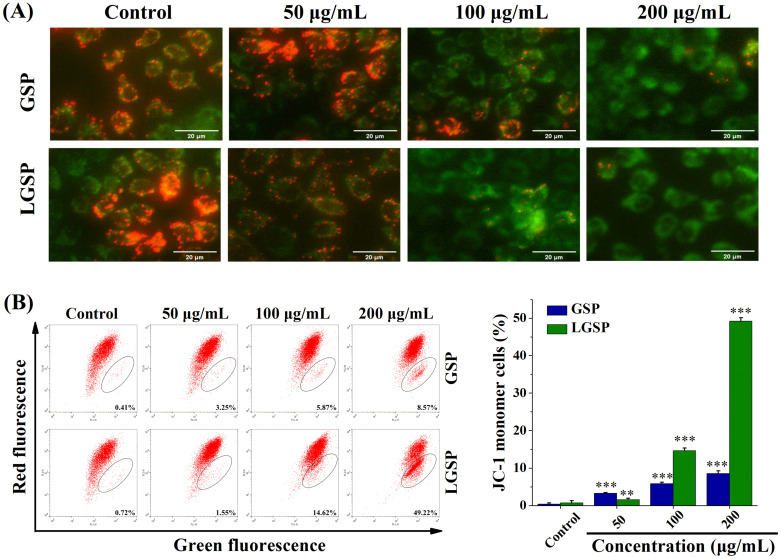
Effect of GSP and LGSP on the mitochondrial membrane potential (MMP) of HeLa cells. HeLa cells were treated with GSP and LGSP for 48 h, and the MMP was (**A**) observed under an inverted fluorescence microscope and (**B**) determined by flow cytometer with JC-1 dye. Data were mean ± SD (*n* = 3): ** 0.001 < *p* < 0.01; *** *p* < 0.001.

**Figure 5 antioxidants-11-00422-f005:**
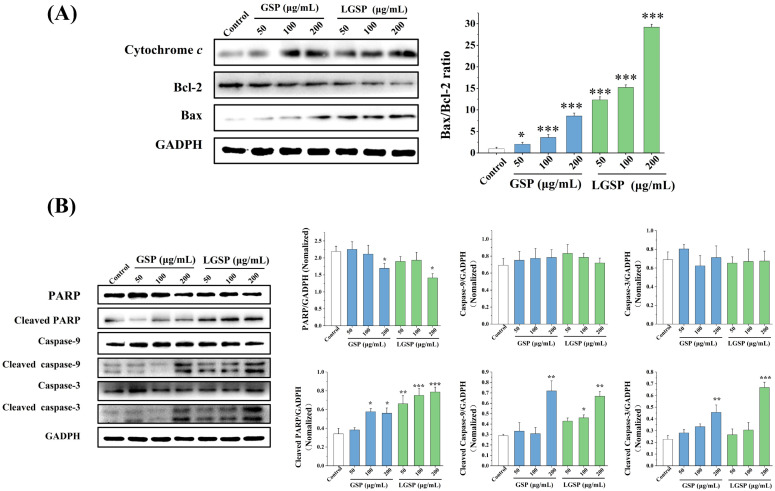
Western blot analysis of the expressions of (**A**) Bax, Bcl-2, cytochrome *c*, (**B**) PARP, cleaved PARP, caspase-9, cleaved caspase-9, caspase-3 and cleaved caspase-3 in HeLa cells treated with different concentrations of GSP or LGSP for 48 h. Data were mean ± SD (*n* = 3): * 0.01 < *p* < 0.05; ** 0.001 < *p* < 0.01; *** *p* < 0.001.

**Figure 6 antioxidants-11-00422-f006:**
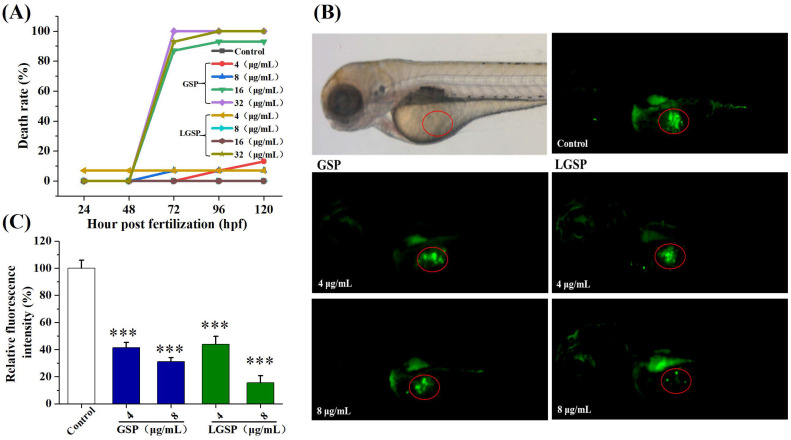
Effect of GSP and LGSP on the HeLa-derived xenograft tumor growth. (**A**) Mortality curve of zebrafish embryos treated with different doses of LGSP and GSP. (**B**) Tumor growth in HeLa-derived xenograft zebrafish embryos treated with GSP and LGSP for 48 h was observed under a microscope. (**C**) The fluorescence intensity of the tumor in Figure 6B. Data were mean ± SD (*n* = 3): *** *p* < 0.001.

## Data Availability

The data presented in this study are available in this manuscript.

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
