# Peer review of "Lipophilic Grape Seed Proanthocyanidin Exerts Anti-Cervical Cancer Effects in HeLa Cells and a HeLa-Derived Xenograft Zebrafish Model"

_antioxidants, 2022, doi:10.3390/antiox11020422_

Round 1

Reviewer 1 Report

The manuscript report the results of Lipophilic grape seed proanthocyanidin (LGSP) on the prevention and treatment of Cervical Cancer.

The study is interesting and the results are clearly presented.

I have two comments and concerns:

In 2.2. Preparation of LGSP,

(The composition of LGSP, shown in Supplementary Figure S1, are mono-, di- and tri-lau-roylated GSP.)

This statement, and the contents of Figure S1 are too general and therefore the molecular composition of the used LGSP should be quantified, otherwise the vagueness of the LGSP composition makes difficult to identify or compare with other polyphenolic compounds.

Some spectroscopic or diffractometric control to check the solid state composition of LGSP and GSP, will be advisable.

5. Conclusions

The conclusions are too general.

The quantitative effects of LGSP on the different biological and biochemical processes studied should be explain in more detail and with the inclusion of the resulted data.

After these comments and concerns are explain and solved, the manuscript may be accepted for publication.

Reviewer 2 Report

This paper studies the Anti-cervical cancer activity of lipophilic grape seed proanthocyanidin, presenting a promising drug for clinical applications. It can be published after addressing the following concerns:
1, It is hard to observe the cell distribution in the bright field image in Fig. 1C before zooming in. Can the authors improve the image contrast for clear observation? In addition, cell counting can be performed to compare the cell density directly. Much open-source software, such as ImageJ-TrackMate, can do this.
2, In Fig. 1C, the cell density looks lower after drug treatment, but the cell density looks unchanged with and without drugs in Fig. 4A. This may lead to some confusion. What is the difference between the experimental conditions for these two figures?
